# Nonlinear Dynamics of Heart Rate Variability after Acutely Induced Myocardial Ischemia by Percutaneous Transluminal Coronary Angioplasty

**DOI:** 10.3390/e25030469

**Published:** 2023-03-08

**Authors:** Martín Calderón-Juárez, Itayetzin Beurini Cruz-Vega, Gertrudis Hortensia González-Gómez, Claudia Lerma

**Affiliations:** 1Plan de Estudios Combinados en Medicina, Faculty of Medicine, Universidad Nacional Autónoma de México, Mexico City 04510, Mexico; martin.cal.j@comunidad.unam.mx (M.C.-J.); ibcruzve@comunidad.unam.mx (I.B.C.-V.); 2Department of Electromechanical Instrumentation, Instituto Nacional de Cardiología Ignacio Chávez, Mexico City 04480, Mexico; 3Department of Physics, Faculty of Sciences, Universidad Nacional Autónoma de México, Mexico City 04510, Mexico; hortecgg@ciencias.unam.mx

**Keywords:** heart rate variability, recurrence plot, surrogate data, nonlinearity, percutaneous transluminal coronary angiography, reperfusion

## Abstract

Several heart rate variability (HRV) characteristics of patients with myocardial ischemia are associated with a higher mortality risk. However, the immediate effect of acute ischemia on the HRV nonlinear dynamical behavior is unknown. The objective of this work is to explore the presence of nonlinearity through surrogate data testing and describe the dynamical behavior of HRV in acutely induced ischemia by percutaneous transluminal coronary angioplasty (PTCA) with linear and recurrence quantification analysis (RQA). Short-term electrocardiographic recordings from 68 patients before and after being treated with elective PTCA were selected from a publicly available database. The presence of nonlinear behavior was confirmed by determinism and laminarity in a relevant proportion of HRV time series, in up to 29.4% during baseline conditions and 30.9% after PTCA without statistical difference between these scenarios. After PTCA, the mean value and standard deviation of HRV time series decreased, while determinism and laminarity values increased. Here, the diminishment in overall variability caused by PTCA is not accompanied by a change in nonlinearity detection. Therefore, the presence of nonlinear behavior in HRV time series is not necessarily in agreement with the change of traditional and RQA measures.

## 1. Introduction

Heart rate variability (HRV) has proven to be a sensitive tool for risk stratification in patients after acute myocardial infarction (AMI) [1]. In general, decreased time-domain indices are related to higher cardiovascular and all-cause mortality [1,2]. More recently, nonlinear measures have been considered to identify high-risk patients [3]. The nonlinear approach, in addition to clinical applications [4], has been implemented to study cardiovascular and autonomic physiology in experimental settings [5].

Nonlinear behavior is often assumed in physiological time series, such as HRV. However, it is appropriate to investigate whether the nonlinear properties at issue are justified by the data [6], as this has implications in the use of methods from the nonlinear approach, particularly in the interpretation of the results. Often, a diminishment in nonlinear measures through aging and pathologic conditions is taken as an indicator of “loss” in nonlinear behavior, although this does not mean that nonlinear dynamic cease to exist in the time series [7].This issue was addressed with Fourier transform-based surrogate data in patients about ten days after AMI [8], in which nonlinear dynamics were identified in HRV using entropy-based measures of time series complexity and regularity.

One of the major drawbacks of Fourier transform-surrogate data testing is that nonlinearity will be detected because the time series are either nonlinear or nonstationary [9]. To overcome this issue, it has been proposed to use methods based on wavelet decomposition to generate surrogate data that preserve nonstationary behavior of time series [10,11]. We have implemented such methods in short-term HRV of healthy subjects and in pathologic conditions [12], in which we propose to study nonlinear behavior through recurrence quantification analysis (RQA). This approach is suitable for short, noisy, and nonstationary time series [13], properties that are present in HRV.

As mentioned above, nonlinearity has been assessed several days after AMI [8], and during the myocardial ischemia [14,15]. Furthermore, revascularization therapy carried out by percutaneous transluminal coronary angioplasty (PTCA) may induce relevant changes in HRV dynamics. HRV linear modifications have been suggested as indicators for improved autonomic regulation during PTCA [16] and during follow-up after PTCA [17,18,19]. However, the HRV dynamics and the presence of nonlinearity have not been assessed by the RQA approach in patients with acutely induced occlusion of major coronary arteries.

The aim of the study is to assess the presence of nonlinearity in HRV time series from patients during acutely induced myocardial ischemia by PTCA, as well as evaluate HRV through traditional and RQA measures.

## 2. Materials and Methods

### 2.1. Study Design and Data Collection

Data collection was obtained from the “STAFF III database” on Physionet [20,21], which includes data from 104 patients who were elective patients for percutaneous coronary intervention (PCI) where single prolonged balloon inflation was introduced in one of the major coronary arteries according to the case. This database was obtained between 1995–1996 at Charleston Area Medical Center (Charleston, WV, USA), containing standard 12-lead ECG recordings.

ECGs in two different moments (lead V) were obtained during the PCI before and after the intervention in two established places: the patient’s room and the catheterization laboratory. For this work pre-inflation (baseline) recordings were obtained immediately before any catheter insertion, and the post-inflation ECG recordings were obtained immediately after the balloon deflation. In both moments, the ECGs were acquired for 5 min during supine position in the catheterization laboratory.

According to database information, there were 20 patients with prior myocardial infarction (MI). To minimize the artifacts and noise from skeletal muscle, the Mason-Likar electrode configuration was used. Data acquisition was performed with a custom-made equipment by Siemens-Elema AB (Solna, Sweden). The ECG was digitized at a sample rate of 1000 Hz with 0.625 μV of amplitude resolution. The study protocol was approved by the Research and Ethics Committee of our institution (protocol number 22–1309).

From the initial sample of 104 patients, during signal processing, QRS detection, and RR beat correction, 36 patients were excluded due to having total recording time < 300 s (n = 1), ECG artifacts, arrhythmias, or sustained ectopic beats (n = 11), not having either the pre-inflation or post-inflation recording in the catheterization laboratory room (n = 12), reading errors in ECG data (n = 7), or having >5% of RR intervals replaced due to arrhythmias with adaptive filtering procedure [22] (n = 5). Finally, 68 subjects were considered for the analysis (40% were female patients). Table 1 shows the clinical characteristics of the studied sample. When comparing the subgroup by sex, both age and proportion of patients having prior MI were similar between groups. All patients with prior MI from the initial sample were included.

### 2.2. ECG Processing and QRS Complex Identification

The ECG recordings of 5 min length were visually inspected by three trained observers. Additionally, correct identification of R waves was visually supervised in the collaborative platform PhysioZoo V1.5.6 software under peak detector “rqrs” allowing manual correction of miss-detections, artifacts, and arrhythmias [23]. An adaptive filter was applied to correct ectopic heartbeats replacing them by normal RR interval values [22]. Heartbeat replacement was applied in 39 recordings pre- (71.4%) and post-inflation (28.6%), considering beat correction of 0.55% as the limit to include. Hence, we refer to NN intervals as the time between “normal” heartbeats [24].

### 2.3. Heart Rate Variability Analysis

#### 2.3.1. Time-Domain and Frequency-Domain Measures

The mean value of 5-min HRV time series was calculated (meanNN [ms]), its standard deviation (SDNN [ms]), and the standard deviation of the difference between consecutive NN intervals (SDSD [ms]) [24]. Before estimation of the frequency-domain measures, NN time series were resampled at 3 Hz, then a 300-data point. Hamming window was applied through the resampled data with 50% overlap. Finally, discrete Fourier transform was applied, and we calculated the low (*LF* (ms^2^), 0.04–0.15 Hz) and high frequency (*HF* (ms^2^), 0.15–0.4 Hz) bands. These measures are also expressed as normalized units (n.u.), as follows [24]:(1)$$LF\left(n.u.\right)=\frac{LF}{Total\;power-VLF},$$
(2)$$HF\left(n.u.\right)=\frac{HF}{Total\;power-VLF},$$
where *VLF* is the very low frequency band (<0.04 Hz). We also show the *LF* (ms^2^)/*HF* (ms^2^) ratio.

#### 2.3.2. Recurrence Quantification Analysis

The embedding parameters for each time series: embedding dimension (m) and embedding delay (τ) were determined by the first local minimum at zero in the false nearest neighbors function and the first local minimum at averaged mutual information function, respectively.

After embedding, the recurrence plot construction is defined in [13]:(3)$$R_{i,j}=\Theta \left(\varepsilon_{i}-\|\vec{x_{i}}-\vec{x_{j}}\|\right),\;i,j=1,\ldots ,N,$$
where a tolerance to define a recurrence $\varepsilon_{i}$ of the point $x_{j}$ in the vicinity centered in $x_{i}$ corresponds, in this case, to the fixed amount of neighbors (FAN) norm (recurrence density was set to = 0.07). If the distance between two points falls within the vicinity, the Heaviside function Θ(x) assigns $R_{i,j}=1$, and said point is considered a recurrence point, otherwise $R_{i,j}=0$. Then a matrix of these binarized values is created and is graphically represented as recurrence plot, where black points correspond to $R_{i,j}=1$, and white points to $R_{i,j}=0$.

The MATLAB toolbox “Cross Recurrence Plot” developed by Marwan et al. [25] (https://tocsy.pik-potsdam.de/CRPtoolbox/, accessed on 29 January 2023) was used to estimate the nonlinear measures determinism (DET) and laminarity (LAM), both measures useful to characterize the autonomic control of HRV in humans, i.e., larger DET and LAM values reflect vagal withdrawal [26]. DET is the proportion of recurrence points that form diagonal structures (Equation (4)) and LAM is the proportion of recurrence points that form vertical structures (Equation (5)), calculated with minimal diagonal $l_{min}=2$ and vertical length $v_{min}=2$, based on the histogram P(l) of diagonal and vertical structures, respectively.
(4)$$DET=\frac{\sum_{l=l_{min}}^{N}lP\left(l\right)}{\sum_{l=1}^{N}lP\left(l\right)}\;$$
(5)$$LAM=\frac{\sum_{v=v_{min}}^{N}lP\left(l\right)}{\sum_{v=1}^{N}lP\left(l\right)}\;$$

### 2.4. Surrogate Data Testing

We applied the algorithm pinned wavelet iterative amplitude adjusted Fourier transform (PWIAAFT) [10] developed by Keylock, C. (https://sites.google.com/site/chriskeylocknet/software, accessed on 29 January 2023), which preserves nonstationary behavior, opposed to Fourier transform-based surrogates that introduce stationarity on surrogate data [9] and may be prone to mistake nonstationarity for nonlinearity. Briefly, in this method, maximal overlap discrete wavelet transform (MODWT) is used to decompose the original time series, then a threshold (ρ) is established to fix a set of wavelet coefficients. On the remaining wavelet scales, the iterative amplitude adjusted Fourier transform (IAAFT) [9,11,27] is applied. Additionally, the algorithm fits a cubic Hermitian polynomial in the fixed coefficients before applying IAAFT in these [11], also referred to as gradual wavelet reconstruction. A flow chart describing the IAFFT algorithm is shown in Figure 1 and the wavelet-based algorithm is described with the flow chart of Figure 2, both procedures are described in detail in [11].

This method was used in 5-min HRV time series in a previous work [12] where no benefit was found from using a threshold for wavelet fixation ρ > 0.01. Therefore, in this work we use ρ = 0.01.

A group of 99 surrogates is generated, and a discriminative nonlinear statistic (in this case, LAM and DET) is measured in the set of surrogates and the original time series. If the value measured on the original data lies beyond the fifth percentile of either side of the statistical distribution curve, this time series is classified as nonlinear.

### 2.5. Statistical Analysis

Categorical clinical variables are expressed as absolute frequency (relative frequency, in brackets) and were compared with the Pearson’s chi-square test. Continuous data had normal distribution (Kolmogorov–Smirnov test, *p* > 0.05) and are shown as mean (±standard deviation). Continues data were compared with paired Student’s t-test or analysis of variance (ANOVA) for repeated measures with post-hoc comparisons adjusted by the Bonferroni method, considering prior MI as the comparison factor between groups and measurement before and after PTCA as the comparison factor within groups. The indices LF(ms^2^), HF(ms^2^), and LF/HF were log transformed. We calculated 95% confidence interval for the percentage of nonlinear time series classified as nonlinear (Clopper–Pearson method). The statistical analysis was performed by Statistical Package for the Social Sciences (SPSS) version 21.0 (IBM Corp., Armonk, NY, USA).

## 3. Results

Table 2 shows the results of HRV linear indices in all participants (N = 68). There are statistically significant differences in meanNN, SDNN, SDSD, ln(LF(ms^2^)), and ln(HF(ms^2^)) between the two moments, being smaller after the PTCA procedure compared with the baseline. LF (n.u.), HF (n.u.), and ln (LF/HF) had no significant difference when compared before and after PTCA.

Table 3 shows the analysis of linear HRV indices before and after PTCA in patients grouped by prior MI (N = 20) or no prior MI (N = 48). While meanNN, SDNN, and SDSD decreased significantly after PTCA only in the group that had no history of prior MI, ln(LF(ms^2^)) and ln(HF(ms^2^)) decreased significantly after PTCA in both groups. LF (n.u.), HF (n.u.), and ln(LF/HF) had no significant change after PTCA in both groups. In all linear HRV indices, there were no significant differences between groups (prior MI vs. no prior MI) for both before and after PTCA.

Figure 3 shows an example of the HRV time series and recurrence plot before PTCA (left column). The panels on the right column correspond to an illustrative surrogate time series and recurrence plot that, overall, exhibits a similar texture to the original. Figure 4 shows an example of the original and surrogate HRV time series after PTCA in the same fashion as Figure 3.

Figure 5 shows the percentage of HRV time series classified as nonlinear before and after PTCA with determinism (DET) and laminarity (LAM) as discriminative nonlinear statistics. The percentage of nonlinear HRV time series is as follows: (a) DET, before PTCA 29.4%, after PTCA 30.9%; (b) LAM, before PTCA 26.5%, after PTCA 23.5%. No statistically significant differences were found before and after PTCA with either of the nonlinear measures.

DET and LAM values were significantly higher after PTCA compared with baseline condition (Figure 6); DET (before [0.666 ± 0.164] vs. after [0.739 ± 0.158] PTCA), LAM (before [0.687 ± 0.192] vs. after [0.761 ± 0.165] PTCA). When comparing DET and LAM of patients grouped by history of prior MI, the increase in determinism and laminarity was significant in both groups, and there were no significant differences between groups (both before and after PTCA) (Figure 7). It can be observed in both Figure 6 and Figure 7 that the values in each patient can increase or decrease after PTCA, rather than moving their values in the same direction. In Figure 7, the plots of SDNN and SDSD were added to illustrate that also in linear HRV indices, individual direction of change after PTCA may vary between subjects in both groups, and yet only in the groups that had no prior history of MI the increase in SDNN and SDSD was significant.

## 4. Discussion

In this work, we tested the presence of nonlinear information in HRV time series of patients with acutely induced ischemia during elective PTCA. We show that the overall statistical variation of HRV time series diminishes immediately after PTCA; also, the DET and LAM values are higher after the revascularization procedure.

In patients undergoing PTCA, SDNN diminished during the first hour after the reperfusion procedure and eventually increased beyond their baseline conditions (before PTCA). Moreover, the mean of RR intervals gradually increased during the next 24 h after PTCA [28]. Other authors, who measured SDNN before and 24 h after PTCA found that this value increases; however, they did not observe any relevant change in SDSD [29]. In the present study, we found a decrease in meanNN, SDNN, and SDSD right after the PTCA. These results reflect that the setpoint of HRV (meanNN) decreases in response to revascularization (i.e., mean heart rate increases), and the gross variability of the time series is narrower (SDNN and SDSD). Moreover, lower SDNN in patients treated with PTCA before hospitalization discharge is related with an increased risk of major clinical events (death or readmission for a new AMI) [30]. Also, we found that the effect of decreasing meanNN, SDSD, and SDNN after PTCA was not significant for patients with prior MI. This suggest that the impact of prior MI is reflected in the modification of the intrinsic cardiac properties, such as electrical conductivity and propagation, that give place to more diverse adjustments in regulatory mechanisms that impact heart rate, including the cardiac autonomic modulation [31].

In a study where 15-min ECG recordings were used for HRV analysis, HF (n.u.) decreased after PCI, but not LF (n.u.) or LF/HF ratio, although SDNN also decreased [32], contrary to another study [28], where the HF (ms^2^) increased after PTCA;nevertheless, these studies are not comparable due to the methodological differences in the measurement of HRV indices. However, in our study, regarding frequency–domain measures we found differences only in the LF and HF indices measured on power units (ms^2^) but did not find any significant differences in the LF and HF indices in normalized units (n.u.) or the LF/HF ratio. The decrease in LF (ms^2^) and HF (ms^2^) after PTCA is more likely an effect of the overall decrease in variability (as observed in the time domain indices) which corresponds to an overall decrease in total power. In contrast, lack of significant change in the proportional contributions of each frequency band of interest (LF and HF in normalized units) indicates that, on average, PTCA had no effect on the autonomic cardiac modulation [24].

Nonlinear measures are of clinical interest, as some have been proposed as predictors of mortality in AMI (detrended fluctuation analysis) [33,34]. Complexity in HRV dynamics of patients after AMI have been assessed with sophisticated methods, such as normalized complexity index, information storage, and Gaussian linear contrast, and showed that the detection of nonlinearity by surrogate data testing depends on the nonlinear property that is measured by a particular metric [8]. In the study mentioned above, the percentage of nonlinear HRV time series after AMI (in rest), using normalized complexity index, was approximately 25%, a similar value as in the present study (DET 30.9%, LAM 23.5%, after PTCA). In our study, we found evidence of nonlinearity with a robust algorithm for surrogate data testing with both nonlinear measures, DET and LAM. However, we did not find any differences in the proportion of nonlinear time series after the intervention. It is possible that the short period of time in which the measures were made was not enough for the manifestation of different dynamics in HRV caused by reperfusion therapy, as it has been suggested in a previous work, in which detrended fluctuation analysis was assessed [35].

However, the algorithm in our work preserves nonstationary behavior in surrogates in contrast with IAAFT. Thus, the time series identified as nonlinear in [8] could mean the original time series were nonstationary or nonlinear [9]. By preserving nonlinearity on surrogate data, we reinforce that the identification of nonlinearity does detect only nonlinearity [9,10,36], as is the case of our work.

The decrease in meanNN and SDSD, and the increase in DET and LAM immediately after PTCA suggest that ischemia-reperfusion does have an effect on the HRV dynamics, in which the autonomic response on cardiac tissue may play a role [37], but it is likely that the biological changes within the cardiac tissue, mainly the induction of oxidative stress, are largely responsible for the quick changes in HRV within the first 5 min after reperfusion [38]. However, more biological research is needed to establish a direct link between the effect of ischemia-reperfusion on the cardiac tissue and HRV measures, as well as the potential clinical applications that it may have.

Increased DET implies a more significant influence of previous heartbeats on the subsequent one, that is, a reduced set of variations for the dynamical states that can acquire the system. On the other hand, LAM is related to a longer permanence of the system in a particular state with the quick capability of a sudden change. These adjustments can reflect the cardiovascular system’s reorganization after the PTCA procedure. Furthermore, we observed that despite the discrepancy of change in DET and LAM in some patients (Figure 7), the average increase in these nonlinear HRV indices in response to PTCA was significant regardless of the history of prior MI, which suggest that the PTCA effect of reducing the set of variations for the dynamical states in the cardiovascular system is strong enough to occur even in those patients with prior MI. Interestingly, other works have attempted to study beat-to-beat nonlinear coupling between NN intervals and arterial pressure in patients after AMI [39] and after sympathetic and parasympathetic pharmacological blockade [40], in which the physiological context might be different from acute myocardial ischemia. Wavelet-based surrogate data might be useful to test the performance of the nonlinear coupling approach.

After PTCA, the statistical variability of time series diminished (SDNN and SDSD) and the values of nonlinear measures increased (DET and LAM), although the proportion of nonlinear time series remains similar to baseline conditions. Hence neither variability in time series nor the value of nonlinear statistics is necessarily related to the “lose” or “gain” of nonlinear behavior.

### Limitations

Our study is limited to patients with pathologic conditions and does not have a valid set of comparable healthy subjects to test whether myocardial ischemia is associated with less or more nonlinear time series. Moreover, it is of physiological interest to test the effect of medication and preexisting comorbidities, but the database used for this work lacks extensive clinical information that may be pertinent to link with HRV. These aspects should be considered in future studies to deepen the factors that potentially determine the nonlinear dynamics of HRV. It has been noted in other works that the proportion of HRV nonlinear time series varies when different nonlinear discriminant statistics are used [8,41], as well as the length of the time series may have an important role in nonlinearity detection, these issues are yet to be studied in future works.

## 5. Conclusions

We confirmed through surrogate data testing and RQA analysis that nonlinear dynamics are present in HRV time series of patients with MI in up to 29.4% of them. The proportion of nonlinear time series did not change significantly immediately after PTCA. After the revascularization procedure, determinism and laminarity increased, which may reflect a different arrangement in data, i.e., there is a change in the dynamic behavior of HRV. However, there was no change in the presence of nonlinear dynamics, despite the loss of statistical variance. Therefore, the HRV dynamic behavior relying on traditional and nonlinear RQA measures should be interpreted with caution.

## Figures and Tables

**Figure 1 entropy-25-00469-f001:**
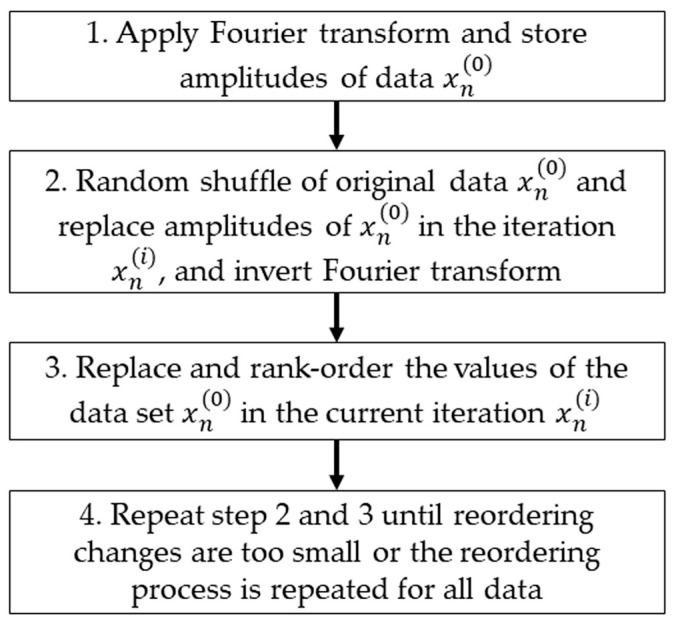
Flow chart of the iterative amplitude adjusted Fourier transform (IAAFT) algorithm [9,11].

**Figure 2 entropy-25-00469-f002:**
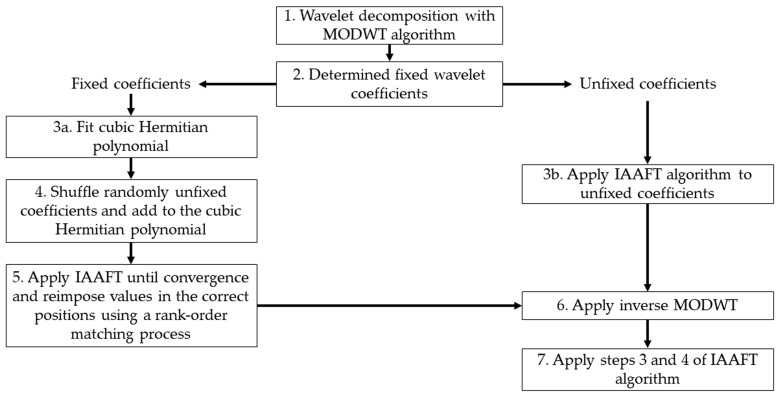
Flow chart of the pinned wavelet iterative amplitude adjusted Fourier transform (PWIAAFT) algorithm (gradual wavelet reconstruction) algorithm. A full description of the algorithm is in [11].

**Figure 3 entropy-25-00469-f003:**
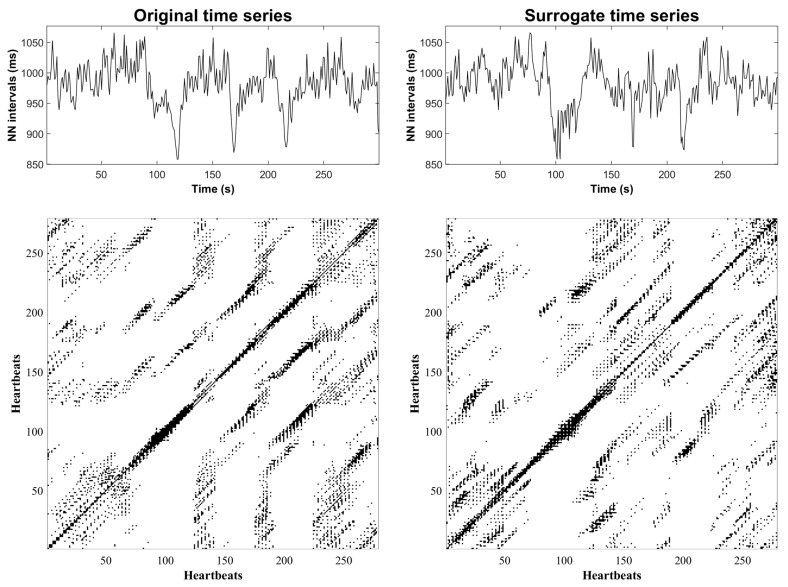
Illustrative examples of an original HRV time series and the corresponding recurrence plot before PTCA (**left** column), and a surrogate HRV time series with the corresponding recurrence plot (**right** column).

**Figure 4 entropy-25-00469-f004:**
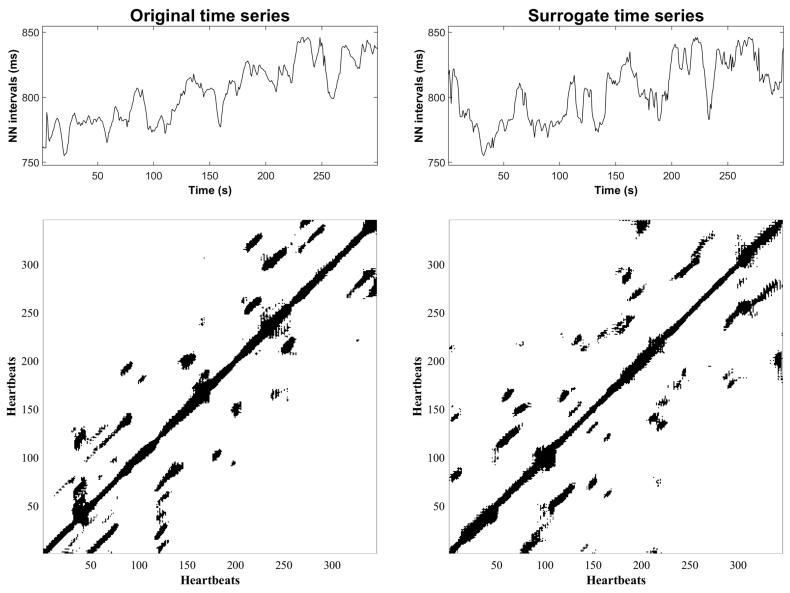
Illustrative example of and original HRV time series and recurrence plot after PTCA (**left** column). Surrogate HRV time series and recurrence plot (**right** column).

**Figure 5 entropy-25-00469-f005:**
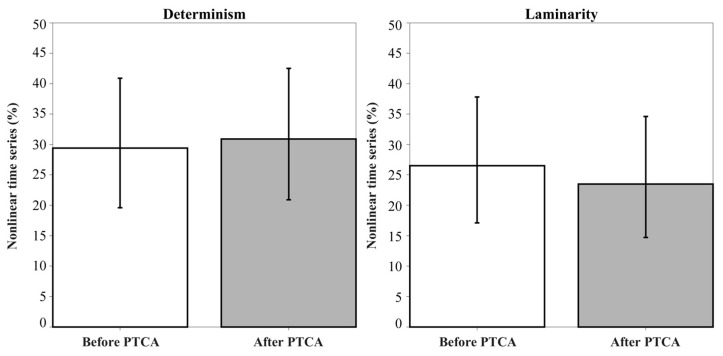
Percentage (95% confidence interval) of nonlinear HRV time series in patients with myocardial infarction (N = 68) before and after PTCA. Determinism (DET) and laminarity (LAM) were used as discriminative nonlinear statistics.

**Figure 6 entropy-25-00469-f006:**
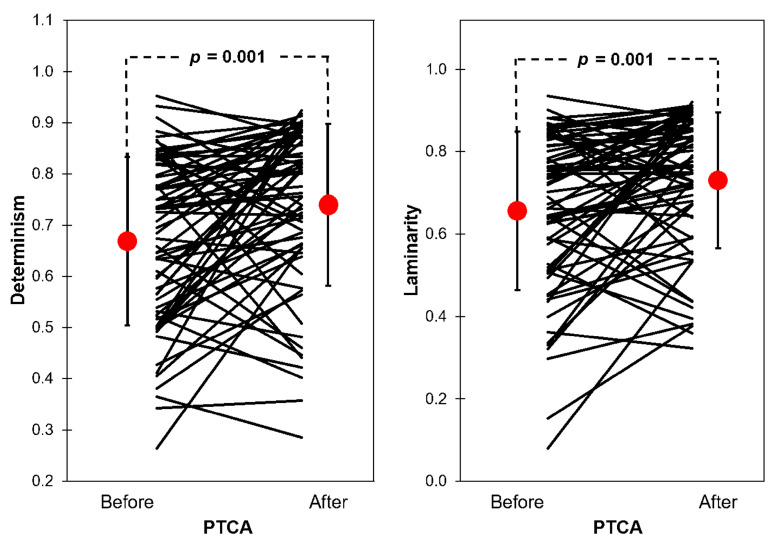
Determinism and laminarity in patients with myocardial infarction (N = 68) before vs. after PTCA, mean (red dot) ± 1 SD is shown.

**Figure 7 entropy-25-00469-f007:**
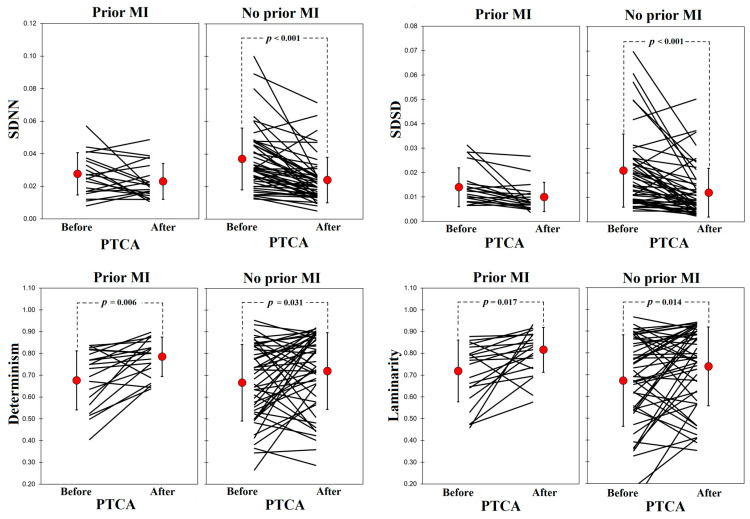
SDNN, SDSD, DET, and LAM in patients before and after PTCA grouped by history of prior myocardial infarction (MI). Mean (red dot) ± 1 SD is shown.

**Table 1 entropy-25-00469-t001:** Clinical characteristics from 68 patients with myocardial ischemia who underwent a percutaneous transluminal coronary angioplasty (PTCA) procedure. Results are shown as mean ± standard deviation or absolute value (percentage).

Variable	All (N = 68)	Female (N = 27)	Male (N = 41)	*p* Value
Age (years)	59 ± 12	59 ± 10	59 ± 13	0.996
Prior MI	20 (29%)	6 (22%)	14 (34%)	0.291

MI: myocardial infarction.

**Table 2 entropy-25-00469-t002:** Heart rate variability indexes from 68 patients with myocardial ischemia who underwent a percutaneous transluminal coronary angiography (PTCA) procedure. Results are shown as mean ± standard deviation.

	Before PTCA	After PTCA	*p* Value
Time-domain measures			
meanNN (ms)	882 ± 149	856 ± 134	0.044
SDNN (ms)	33.9 ± 17.9	23.8 ± 13.1	<0.001
SDSD (ms)	18.8 ± 13.9	11.6 ± 9.3	<0.001
Frequency-domain measures			
ln(LF (ms^2^)) ^1^	5.107 ± 1.358	4.044 ± 1.438	<0.001
ln(HF (ms^2^)) ^1^	3.818 ± 1.397	2.697 ± 1.388	<0.001
LF (n.u.)	74.2 ± 17.8	75.4 ± 17.2	0.580
HF (n.u.)	25.9 ± 17.8	25.1 ± 17.0	0.683
ln(LF/HF) ^1^	1.289 ± 1.084	1.348 ± 1.035	0.660

^1^ Natural logarithm (ln) was used for log transformation. meanNN: mean of NN intervals. SDNN: standard deviation of NN intervals. SDSD: standard deviation of the difference between consecutive NN intervals. LF: low frequency. HF: high frequency.

**Table 3 entropy-25-00469-t003:** Heart rate variability indexes from patients who underwent PTCA grouped by history of prior myocardial infarction (MI). Results are shown as mean ± standard deviation.

	Prior MI (N = 20)		No Prior MI (N = 48)	
	Before PTCA	After PTCA	Before PTCA	After PTCA
meanNN (ms)	864 ± 147	850 ± 130	891 ± 152	860 ± 137 *
SDNN (ms)	27.6 ± 12.8	23.2 ± 10.8	36.5 ± 19.2	24.1 ± 14.0 **
SDSD (ms)	14.2 ± 7.9	9.6 ± 5.6	20.7 ± 15.4	12.4 ± 10.5 **
ln(LF (ms^2^))	4.846 ± 1.358	4.195 ± 1.450 *	5.216 ± 1.357	3.981 ± 1.444 **
ln(HF (ms^2^))	3.462 ± 1.287	2.646 ± 1.153 **	3.966 ± 1.427	2.718 ± 1.485 **
LF (n.u.)	75.783 ± 15.800	77.741 ± 17.568	73.490 ± 18.715	74.399 ± 17.093
HF (n.u.)	24.268 ± 15.832	22.259 ± 17.568	26.510 ± 18.715	26.082 ± 16.874
ln(LF/HF) ^1^	1.385 ± 1.060	1.549 ± 1.136	1.249 ± 1.103	1.264 ± 0.990

^1^ Natural logarithm (ln) was used for log transformation. meanNN: mean of NN intervals. SDNN: standard deviation of NN intervals. SDSD: standard deviation of the difference between consecutive NN intervals. LF: low frequency. HF: high frequency. * *p <* 0.05 (before vs. after PTCA, same group). ** *p* < 0.01 (before vs. after PTCA, same group).

## Data Availability

The data used in this study is publicly available at https://physionet.org/content/staffiii/1.0.0/ (accessed on 29 January 2023).

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
