# Peer review of "Nonlinear Dynamics of Heart Rate Variability after Acutely Induced Myocardial Ischemia by Percutaneous Transluminal Coronary Angioplasty"

_entropy, 2023, doi:10.3390/e25030469_

Round 1

Reviewer 1 Report

In the present study nonlinear behaviour of heart rate variability (HRV)  is investigated in patients undergoing percutaneous transluminal coronary angiography (PTCA), by way of recurrence quantitative analysis. Short term HRV series are considered for 68 patients from a publicly available database before and after PTCA and their linear properties (spectral analysis) and nonlinear properties ( determinism and laminarity) are investigated. The percentage of series presenting nonlinear behavior was similar across conditions but determinism and laminarity as well as mean and standard deviation of HP decreased immediately after PTCA.

The manuscript is generally clear and well organized, but the experimental setup and methods should be described in further detail and some additional discussion of the previous literature with additional references would better elucidate the contribution of the present work to the state of the art. A general revision of the English language would also remove some of the typographical errors present, however the manuscript is generally well written and comprehensible. The statistical analysis, tables and figures are appropriate.

Major comments:

1.      More detail is needed in the Heart rate variability analysis section and in particular a definition of determinism and laminarity should be added. Their physiological meaning and why these specific measures of nonlinearity were chosen should be addressed. It is known that in the field at hand different nonlinear methods can give different results (Porta et al, Front. Physiol. 6, 71 (2015)), how would the agreement between these two measures be interpreted? Is it because they investigate similar time scales? What are the differences between the two?

2.      SDSD should be defined in the appropriate section (currently the first mention of it is in the Results chapter and no definition is given until the Discussion).

3.      Could prior myocardial infarction be a confounding factor at all in the analysis? Should a sub-analysis of prior MI vs no prior MI be performed? The percentage of subjects with prior MI is discussed in the Study design and data collection section but is not further mentioned in the paper, therefore I am wondering how relevant it is.

4.      Lines 237-242: please comment on the potential reasons for the lack of significant differences in the frequency domain measures shown in this study.

5.      While in the Discussion some comparisons with previous works on myocardial infarction is present I believe it could be further expanded, using if relevant: Bai et al, Am. J. Physiol. Heart Circ. Physiol. 295(2), H578–H586 (2008); Nollo et al, Computers in Cardiology 2000, Vol. 27, pp. 143–146; Stein et al, J. Cardiovasc. Electrophysiol. 16(1), 13–20 (2005) and similar works.

Minor comments:

1.      There is some ambiguous phrasing in the Study design and data collection section: while it becomes clearer in the Discussion, please specify in this chapter as well at what time the 5-minute HRV is recorded post-PTCA intervention (immediately after deflation is different than after 1 hour as discussed by the authors). Furthermore, it is unclear if the enrolled patients are 104 (line 66) or 108 (line 84), if excluded patients include any with previous MI (from the table it would seem like they are not, but it should be explicitly reported in the text) and if all procedures are performed in the catheterization laboratory.

2.      Statistical analysis section: please specify what software was used for statistical analysis.

3.      Minor typographical error: lines 39-40 “However, it is appropriate to investigate whether the nonlinear properties at issue are justified by de data [5], as this has implications in the use of methods from the nonlinear approach, particularly in the interpretation of the results”; lines 80-81“Data acquisition was using custom-made equipment by 80 Siemens-Elema AB (Solna, Sweden)”.

4.      Table 2: please add definition of abbreviations in the table caption.

5.      Lines 253-254 sentence referencing [28] is unclear, please elaborate further.

Author Response

Response to Reviewer 1

Comment: In the present study nonlinear behaviour of heart rate variability (HRV)  is investigated in patients undergoing percutaneous transluminal coronary angiography (PTCA), by way of recurrence quantitative analysis. Short term HRV series are considered for 68 patients from a publicly available database before and after PTCA and their linear properties (spectral analysis) and nonlinear properties ( determinism and laminarity) are investigated. The percentage of series presenting nonlinear behavior was similar across conditions but determinism and laminarity as well as mean and standard deviation of HP decreased immediately after PTCA.

The manuscript is generally clear and well organized, but the experimental setup and methods should be described in further detail and some additional discussion of the previous literature with additional references would better elucidate the contribution of the present work to the state of the art. A general revision of the English language would also remove some of the typographical errors present, however the manuscript is generally well written and comprehensible. The statistical analysis, tables and figures are appropriate.

Response: We thank the Reviewer for the positive comments and helpful suggestions, which helped to improve our manuscript. The revised manuscript has more detailed description on several methodological aspects, relevant references were added, and we also corrected typographical errors.

Comment: Major comments:

  1. More detail is needed in the Heart rate variability analysis section and in particular a definition of determinism and laminarity should be added. Their physiological meaning and why these specific measures of nonlinearity were chosen should be addressed. It is known that in the field at hand different nonlinear methods can give different results (Porta et al, Front. Physiol. 6, 71 (2015)), how would the agreement between these two measures be interpreted? Is it because they investigate similar time scales? What are the differences between the two?

Response: As the reviewer points out, the use of different discriminating statistics leads to a different proportion of nonlinear HRV time series (10.1063/1.5115506), this is also observed in the work that the reviewer suggests (10.3389/fphys.2015.00071). We hypothesize from this, that there are various properties contained in the HRV dynamics that are reachable for a given discriminating statistic measure, but not for others. Even in here with recurrence plots measures, determinism (based on diagonal lines) and laminarity (based on vertical lines) focus on distinct dynamic properties: similar recurrence evolution and laminar states and intermittency (10.1016/j.physrep.2006.11.001). It has been discussed by Javorka M et al. (10.1088/0967-3334/30/1/003) from experimentation in animal models and humans that the increase in determinism and laminarity reflects a vagal withdrawal. However, a future perspective for this work is to compare recurrence quantification analysis with other nonlinear measures, as well as comparing different time scales. All the above has been added to the new version of the manuscript (lines 147 to 152; 345 to 348; 372 to 376).

Comment: 2.   SDSD should be defined in the appropriate section (currently the first mention of it is in the Results chapter and no definition is given until the Discussion).

Response: The definition of SDSD is now introduced in the Methods section (lines 120 – 121).

Comment: 3.      Could prior myocardial infarction be a confounding factor at all in the analysis? Should a sub-analysis of prior MI vs no prior MI be performed? The percentage of subjects with prior MI is discussed in the Study design and data collection section but is not further mentioned in the paper, therefore I am wondering how relevant it is.

Response: The impact of prior MI was analyzed, and the results are presented in Table 3 and Figure 7. As described on lines 215 to 221 for the linear HRV indices, while meanNN, SDNN, and SDSD decreased significantly after PTCA only in the group that had no history of prior MI, Ln(LF(ms2)) and Ln(HF(ms2)) decreased significantly after PTCA in both groups. LF(n.u.), HF(n.u.), and ln(LF/HF) had no significant change after PTCA in both groups (Table 3). These findings suggest that the impact of MI is reflected in the modification of the intrinsic cardiac properties, such as electrical conductivity and propagation, that give place to more diverse adjustments in regulatory mechanisms that impact heart rate, including autonomic regulation. This is now mentioned in the Discussion section (lines 309 to 317).

In contrast, the effect of increasing the nonlinear indices (determinism and laminarity) occurred regardless of history of prior MI (Figure 7), which suggest that the PTCA effect of reducing the set of variations for the dynamical states in the cardiovascular system is strong enough to occur even in those patients with prior MI. This is now mentioned on lines 264 to 267.  

Comment: 4.      Lines 237-242: please comment on the potential reasons for the lack of significant differences in the frequency domain measures shown in this study.

Response: Regarding frequency-domain measures we found differences only in the LF and HF indices measured on power units (ms2) but did not find significant differences in the LF and HF indices in normalized units (n.u.) or the LF/HF ratio regarding frequency-domain measures. The decrease on LF(ms2) and HF(ms2) after PTCA is more likely an effect of the overall decrease of variability (as observed in the time domain indices) which corresponds to an overall decrease in total power. In contrast, lack of significant change in the proportional contributions of each frequency band of interest (LF and HF in normalized units) indicates that, on average, PTCA had no effect in the autonomic cardiac modulation. This is now mentioned on lines 309 to 317.

Comment: 5.      While in the Discussion some comparisons with previous works on myocardial infarction is present I believe it could be further expanded, using if relevant: Bai et al, Am. J. Physiol. Heart Circ. Physiol. 295(2), H578–H586 (2008); Nollo et al, Computers in Cardiology 2000, Vol. 27, pp. 143–146; Stein et al, J. Cardiovasc. Electrophysiol. 16(1), 13–20 (2005) and similar works.

Response: We have added the recommended works in either the introduction or discussion sections, and propose that wavelet-based surrogate data testing could be useful to test the performance of nonlinear coupling in HRV (lines 354 to 359).

Comment: Minor comments:

  1. There is some ambiguous phrasing in the Study design and data collection section: while it becomes clearer in the Discussion, please specify in this chapter as well at what time the 5-minute HRV is recorded post-PTCA intervention (immediately after deflation is different than after 1 hour as discussed by the authors). Furthermore, it is unclear if the enrolled patients are 104 (line 66) or 108 (line 84), if excluded patients include any with previous MI (from the table it would seem like they are not, but it should be explicitly reported in the text) and if all procedures are performed in the catheterization laboratory.

Response: We mention now in the study design that the ECG recordings were obtained immediately before balloon inflation and immediately after balloon deflation in the catheterization laboratory (lines 76 to 84). The enrolled patients were 104 and all patients with prior MI from the initial sample were included, as is now mentioned on line XX.

Comment: 2.      Statistical analysis section: please specify what software was used for statistical analysis.

Response: We used SPSS version 21.0, as mentioned now in the Statistical analysis section.

Comment: 3.      Minor typographical error: lines 39-40 “However, it is appropriate to investigate whether the nonlinear properties at issue are justified by de data [5], as this has implications in the use of methods from the nonlinear approach, particularly in the interpretation of the results”; lines 80-81“Data acquisition was using custom-made equipment by 80 Siemens-Elema AB (Solna, Sweden)”.

Response: The typographical errors were corrected.

Comment: 4.      Table 2: please add definition of abbreviations in the table caption.

Response: Done, please see new Tables 2 and 3.

Comment: 5.      Lines 253-254 sentence referencing [28] is unclear, please elaborate further.

Response: This sentence was edited to indicate that previous evidence suggests that the length of the HRV time series influences nonlinear dynamics appreciation (lines 326 to 329).

Reviewer 2 Report

The authors quantify changes in nonlinear indexes of heart rate variability (HRV) from recurrence quantification analysis (RQA) in patients undergoing coronary angiography (PTCA). They report increased determinism and laminarity after the intervention, without changes in the percentage of nonlinear behavior, which was significantly detected in about 30% of the patients.

The results are interesting and the methodology for nonlinear analysis appears appropriate. However, several major methodological issues should be addressed.

1) Population. The characterization of the patients' population is poor and table 1 should report more pieces of information on factors that may influence the HRV dynamics, for instance, body mass index, prevalence of diabetes or hypertension, and taking medications (e.g., beta-blockers).

2) Frequency-domain measures. In the discussion section, the authors conclude that "we did not find significant differences regarding frequency-domain measures" (line 241) but they did not measure frequency domain indexes apart from the sympatho/vagal balance! Table 1 should report the HF power, in ms^2, i.e., the frequency-domain vagal index (as recommended by the cited Guidelines, ref. 19), and not the normalized HF power, which is an index of vago/sympathetic balance. It seems that the authors normalized the LF power and the HF power by the sum of the LF and HF powers, so that HFnu=1-LFnu. This means that 1) HFnu duplicates the information of LFnu, and there are no reasons to include it in the table, and 2) the definition of HFnu is not the same as used in the work of Seetharam et al (2022), ref. 25, where the normalization included total and VLF powers. Therefore the present results cannot be compared with the previous literature as the authors did at lines 236-8 of the discussion. Table 2 should include and compare VLF, LF, and HF powers (in ms^2) to properly characterize the frequency-domain measures.

3) Non-linear measures. Why do the authors report determinism and laminarity in Figure 4 when the more logical way to present them is by adding two new rows in Table 2, under a section on complexity-domain measures? The table can provide the readers with precise numerical values of mean and SD for determinism and laminarity. By the way, the legend of Figure 4 does not indicate whether the vertical axes are SD, SEM, or other measures of dispersion.

4) Commenting on Figure 4, the authors highlight that "the values in each patient can increase or decrease after PTCA, rather than moving their values in the same direction" (line 214). I find this pattern rather surprising considering that the significance of the paired-t test is very low (p=0.001), suggesting much more similar changes after PTCA among the patients. In any case, time-domain measures have a similarly low statistical significance (around 0.001, Table 2) and we might expect similar nonunivocal changes for SDNN and SDSD too. Therefore, the authors should consider replacing Figure 4 with a new figure to be presented in the Supplementary Data. This new figure should include not only the determinism and laminarity panels now in Figure 4, but also the SDNN and SDSD panels to show the individual 68 changes from "before" to "after" PTCA for both time-domain and complexity-domain indexes.

5) The authors applied an interesting surrogate data test (PWIAAFT) that preserves nonstationarity. Unlike IAAFT, this method is relatively new and the authors should describe it in much more detail. For instance, how did they establish the threshold rho? What are the "remaining scales" (line 143)? Are those with tau shorter than rho? What did the authors mean by writing that they fit a polynomial  "at the beginning of the routine"? That the surrogate analysis is applied to the detrended series? A block diagram that represents the algorithm in detail can make the methodology much clearer.

6) It is also unclear what is represented by the vertical axes of Figure 3, which reports the percentage of time series which did not pass the linearity test. As lines 202 and 203 report (duplicating the figure), a percentage is a single number without an associated dispersion. Please explain.

7) At the end of the abstract and at the end of the conclusion section the authors wrote that "the interpretation ... relying on traditional and nonlinear RQA measures should be provided with caution" because they observed a decrease of variance without changes in the percentage of the recordings where nonlinearity was statistically detected. This point should be better discussed. I do not understand why a decrease in the overall variability should imply changes in nonlinearity. For linear (e.g., white noise) and  chaotic series the results of the surrogate test do not change if the series are multiplied by 0.1 or 10, while the variance of the series varies dramatically.  Why in such cases "the interpretation should be provided with caution"?

Minor.

line 225. The text defines "percutaneous transluminal coronary angioplasty (PTCAp)"; but the revascularization therapy was defined as "percutaneous coronary transluminal angiography (PTCA)" at line 55. What is the difference between PTCAp and PTCA?

line 260. "by preserving nonlinearity on surrogate data". Do the authors mean "by preserving nonstationarity on surrogate data"?

lines 280-286. These lines should appear in a separate "Limitations" paragraph.

It seems that RQA should be the abbreviation of Recurrence Quantification Analysis, not Recurrence Quantitative Analysis.

Author Response

Response to Reviewer 2

Comment: The authors quantify changes in nonlinear indexes of heart rate variability (HRV) from recurrence quantification analysis (RQA) in patients undergoing coronary angiography (PTCA). They report increased determinism and laminarity after the intervention, without changes in the percentage of nonlinear behavior, which was significantly detected in about 30% of the patients.

The results are interesting and the methodology for nonlinear analysis appears appropriate. However, several major methodological issues should be addressed.

1) Population. The characterization of the patients' population is poor and table 1 should report more pieces of information on factors that may influence the HRV dynamics, for instance, body mass index, prevalence of diabetes or hypertension, and taking medications (e.g., beta-blockers).

Response: Unfortunately, in the public database used for the present work, the only available information about the patients is the one included in the manuscript. This is mentioned in the study limitations (lines 368 to 372).

Comment: 2) Frequency-domain measures. In the discussion section, the authors conclude that "we did not find significant differences regarding frequency-domain measures" (line 241) but they did not measure frequency domain indexes apart from the sympatho/vagal balance! Table 1 should report the HF power, in ms^2, i.e., the frequency-domain vagal index (as recommended by the cited Guidelines, ref. 19), and not the normalized HF power, which is an index of vago/sympathetic balance. It seems that the authors normalized the LF power and the HF power by the sum of the LF and HF powers, so that HFnu=1-LFnu. This means that 1) HFnu duplicates the information of LFnu, and there are no reasons to include it in the table, and 2) the definition of HFnu is not the same as used in the work of Seetharam et al (2022), ref. 25, where the normalization included total and VLF powers. Therefore, the present results cannot be compared with the previous literature as the authors did at lines 236-8 of the discussion. Table 2 should include and compare VLF, LF, and HF powers (in ms^2) to properly characterize the frequency-domain measures.

Response: As suggested by the reviewer and the cited guidelines, we show the calculation of unnormalized units for LF and HF (ms2). The computations of these metrics (n.u.) are now explicitly defined (Eqs. 2 and 3) and comply with the guidelines used in this work. However, the 5-minutes HRV time series do not have an appropriate length to accurately estimate VLF (10.1161/01.CIR.93.5.1043), hence we limited the report of our results to LF and HF bands. Note that we updated the measurement units to ms, in order to be consistent with the reported units of LF (ms2) and HF (ms2).

Comment: 3) Non-linear measures. Why do the authors report determinism and laminarity in Figure 4 when the more logical way to present them is by adding two new rows in Table 2, under a section on complexity-domain measures? The table can provide the readers with precise numerical values of mean and SD for determinism and laminarity. By the way, the legend of Figure 4 does not indicate whether the vertical axes are SD, SEM, or other measures of dispersion.

Response: We show traditional HRV measures to provide a common background for reference with other available works and as a first general approach to the time series. To emphasize the RQA analysis as a new contribution of this work, we decided to visually present RQA measures to highlight it to the readers. The exact values of the mean and SD are now included in the text description (lines 263 to 264), also we clarify that we are reporting SD in the legend of Figure 6.

Perhaps, one of the most common RQA measures in recurrence plots is DET, since it was an early measure for the quantification of textures in recurrence plots (10.1152/jappl.1994.76.2.965). LAM was presented as a potentially useful complexity measure in HRV time series (10.1103/PhysRevE.66.026702). Javorka M (10.1088/0967-3334/30/1/003) shows that the increased values of these RQA measures reflect a vagal withdrawal, based on experimental observations in animal models and humans (lines 147 to 149). Indeed, this work can be extended to more measures of complexity (even outside recurrence plot measure) and is a future perspective of this research (lines 372 to 375).

Comment: 4) Commenting on Figure 4, the authors highlight that "the values in each patient can increase or decrease after PTCA, rather than moving their values in the same direction" (line 214). I find this pattern rather surprising considering that the significance of the paired-t test is very low (p=0.001), suggesting much more similar changes after PTCA among the patients. In any case, time-domain measures have a similarly low statistical significance (around 0.001, Table 2) and we might expect similar nonunivocal changes for SDNN and SDSD too. Therefore, the authors should consider replacing Figure 4 with a new figure to be presented in the Supplementary Data. This new figure should include not only the determinism and laminarity panels now in Figure 4, but also the SDNN and SDSD panels to show the individual 68 changes from "before" to "after" PTCA for both time-domain and complexity-domain indexes.

Response: The figures showing the direction of change of all individual patients illustrate that despite the discrepancy in the response to PTCA of some patients, the average response in the group is consistent with a strong effect of PTCA, i.e., the paired t-test can identify a direction of change in response to PTCA despite the large heterogeneity in baseline values between patients. In the revised manuscript we describe both the average and individual results of determinism, laminarity, SDNN and SDSD with a new figure where the analysis also considers the effect of prior MI (Figure 7, lines 264 to 273, and Discussion section, lines 299 to 303).

Comment: 5) The authors applied an interesting surrogate data test (PWIAAFT) that preserves nonstationarity. Unlike IAAFT, this method is relatively new and the authors should describe it in much more detail. For instance, how did they establish the threshold rho? What are the "remaining scales" (line 143)? Are those with tau shorter than rho? What did the authors mean by writing that they fit a polynomial  "at the beginning of the routine"? That the surrogate analysis is applied to the detrended series? A block diagram that represents the algorithm in detail can make the methodology much clearer.

Response: In the new version of the manuscript we included all the requested details (lines 164 to 169). Briefly, the threshold rho was determined for 5-minutes HRV time series in a previous reference [10], in which we did not find a benefit to increase the value of the threshold and pin higher energy levels in maximum overlap wavelet transformation (MODT) (10.3389/fphys.2022.807250). The details about the used algorithm are reported by C. Keylock (10.5194/npg-17-615-2010) and following the reviewer’s suggestion we show it in the revised manuscript in two block diagrams (Figures 1 and 2).

Comment: 6) It is also unclear what is represented by the vertical axes of Figure 3, which reports the percentage of time series which did not pass the linearity test. As lines 202 and 203 report (duplicating the figure), a percentage is a single number without an associated dispersion. Please explain.

Response: We report the exact numeric values of the percentage of HRV time series classified as nonlinear, for further reference. As one of the main outcomes of this work, we show it in a figure to visually present that the values are very similar. In this respect we report the confidence interval for a proportion (Clopper-Pearson method).

Comment: 7) At the end of the abstract and at the end of the conclusion section the authors wrote that "the interpretation ... relying on traditional and nonlinear RQA measures should be provided with caution" because they observed a decrease of variance without changes in the percentage of the recordings where nonlinearity was statistically detected. This point should be better discussed. I do not understand why a decrease in the overall variability should imply changes in nonlinearity. For linear (e.g., white noise) and  chaotic series the results of the surrogate test do not change if the series are multiplied by 0.1 or 10, while the variance of the series varies dramatically.  Why in such cases "the interpretation should be provided with caution"?

Response: Indeed, there is not a clear reason why a diminished value of a given nonlinear value should be associated with the existence of nonlinear behavior, but it  is a common misconception in the field. For example, in healthy aging, diminished nonlinear metrics and narrower variability are considered a hint for “less” complexity or nonlinearity (10.1001/jama.1992.03480130122036, 10.1088/0967-3334/33/8/1289, 10.1126/sageke.2004.16.pe16). Thus, we start our approach from that point and accordingly show through our results that the above-mentioned statement is not (always) true. However, we take this opportunity to better explain in our paper the contribution of our work to this issue (lines 25 to 27; 42 to 45).

Comment: Minor.

line 225. The text defines "percutaneous transluminal coronary angioplasty (PTCAp)"; but the revascularization therapy was defined as "percutaneous coronary transluminal angiography (PTCA)" at line 55. What is the difference between PTCAp and PTCA?

Response: After getting advice from a clinical expert, we learned that currently is defined “coronary angiograpy” to all procedures that involve only diagnosis, while “percutaneous transluminal coronary angioplasty” or PTCA, refers to all procedures where the patient receives also treatment with balloon inflation. Therefore, we corrected the term PTCA throughout the revised manuscript accordingly. A similar work that used the same database also referred the term “percutaneous transluminal coronary angioplasty” as PTCA (10.1016/j.medengphy.2008.12.006).

Comment: lines 280-286. These lines should appear in a separate "Limitations" paragraph.

Response: Agreed

Comment: It seems that RQA should be the abbreviation of Recurrence Quantification Analysis, not Recurrence Quantitative Analysis.

Response: Agree

Round 2

Reviewer 1 Report

Authors have answered all my questions and I believe the manuscript has significantly improved.

Author Response

Manuscript ID: entropy-2219806 “Nonlinear dynamics of heart rate variability after acutely induced myocardial ischemia by percutaneous transluminal coronary angiography”

Response to Reviewer 1

Comment: Authors have answered all my questions and I believe the manuscript has significantly improved.

Response: We thank the reviewer’s suggestions which helped to improve the manuscript.

Reviewer 2 Report

The authors addressed properly my previous remarks and the overall quality of the manuscript improved substantially. However, the authors should check the data reported for LF/HF and LFnu indices in Table 3. It is very unlikely that these values coincide with the precision of 3 decimals in the two groups.

Minor. Line 404. The abbreviation DFA is not defined

Author Response

Manuscript ID: entropy-2219806 “Nonlinear dynamics of heart rate variability after acutely induced myocardial ischemia by percutaneous transluminal coronary angioplasty”

Response to Reviewer 2

Comment: The authors addressed properly my previous remarks and the overall quality of the manuscript improved substantially. However, the authors should check the data reported for LF/HF and LFnu indices in Table 3. It is very unlikely that these values coincide with the precision of 3 decimals in the two groups.

Response: We thank the reviewer for this observation. After double-checking all the data, we identified and corrected several errors that occurred during the transcription of the results into Table 3. We verified that they were only transcription errors as all statistical analysis in the original manuscript were correct, and therefore the description of the results and its interpretations is the same in the revised manuscript.

Comment: Minor. Line 404. The abbreviation DFA is not defined.

Response: We replaced the abbreviation with the full name (detrended fluctuation analysis).